# E-Cadherin, Integrin Alpha2 (Cd49b), and Transferrin Receptor-1 (Tfr1) Are Promising Immunohistochemical Markers of Selected Adverse Pathological Features in Patients Treated with Radical Prostatectomy

**DOI:** 10.3390/jcm10235587

**Published:** 2021-11-27

**Authors:** Piotr Zapała, Łukasz Fus, Zbigniew Lewandowski, Karolina Garbas, Łukasz Zapała, Barbara Górnicka, Piotr Radziszewski

**Affiliations:** 1Department of General, Oncological and Functional Urology, Medical University of Warsaw, 02-091 Warsaw, Poland; zapala.piotrek@gmail.com (P.Z.); kgarbas98@gmail.com (K.G.); lzapala@wum.edu.pl (Ł.Z.); pradziszewski@wum.edu.pl (P.R.); 2Department of Pathology, Medical University of Warsaw, 02-091 Warsaw, Poland; barbara.gornicka@wum.edu.pl; 3Department of Epidemiology and Biostatistics, Medical University of Warsaw, 02-091 Warsaw, Poland; zbigniew.lewandowski@wum.edu.pl

**Keywords:** CD49b, e-cadherin, heregulin, NF2, PTEN, TfR1, extraprostatic extension of prostate cancer, nodal involvement of prostate cancer, PSA, nomogram

## Abstract

In patients treated for prostate cancer (PCa) with radical prostatectomy (RP), determining the risk of extraprostatic extension (EPE) and nodal involvement (NI) remains crucial for planning nerve-sparing and extended lymphadenectomy. The study aimed to determine proteins that could serve as immunohistochemical markers of locally advanced PCa. To select candidate proteins associated with adverse pathologic features (APF) reverse-phase protein array data of 498 patients was retrieved from The Cancer Genome Atlas. The analysis yielded 6 proteins which were then validated as predictors of APF utilizing immunohistochemistry in a randomly selected retrospective cohort of 53 patients. For univariate and multivariate analysis, logistic regression was used. Positive expression of TfR1 (OR 13.74; *p* = 0.015), reduced expression of CD49b (OR 10.15; *p* = 0.013), and PSA (OR 1.29; *p* = 0.013) constituted independent predictors of EPE, whereas reduced expression of e-cadherin (OR 10.22; *p* = 0.005), reduced expression of CD49b (OR 24.44; *p* = 0.017), and PSA (OR 1.18; *p* = 0.002) were independently associated with NI. Both models achieved high discrimination (AUROC 0.879 and 0.888, respectively). Immunohistochemistry constitutes a straightforward tool that might be easily utilized before RP. Expression of TfR1 and CD49b is associated with EPE, whereas expression of e-cadherin and CD49b is associated with NI. Since following immunohistochemical markers predicts respective APFs independently from PSA, in the future they might supplement existing preoperative nomograms or be implemented in novel tools.

## 1. Introduction

Along with advances in surgical technique, salvage therapies, and in the perspective of emerging neoadjuvant options, radical prostatectomy (RP) is being constantly confirmed as the fundament of multimodal treatment for locally advanced prostate cancer [1]. In patients treated with radical prostatectomy, preoperative prediction of pN1 and pT3 disease remains crucial for lymphadenectomy [2,3] and nerve-sparing [4] issues, respectively. The pathophysiology of the local and nodal spread of prostate cancer (PCa) remains poorly understood. A plethora of genes’ rearrangements affecting protein expression and impacting the microenvironment have been postulated as predictors of rapid progression or promotors of extraprostatic extension (EPE) and nodal involvement (NI) to date. Nevertheless, none of these has been considered a clinically valid marker, which can aid decision making.

The study aimed to determine protein markers of NI that could potentially be used as immunohistochemical markers of locally advanced prostate cancer.

## 2. Materials and Methods

### 2.1. Candidate Protein Selection

To select candidate proteins associated with adverse pathologic features for clinical validation, The Cancer Genome Atlas (https://portal.gdc.cancer.gov/, accessd on 24 November 2021) (TCGA) was used. Reverse-phase protein array (RPPA) data of 498 patients treated with RP and extended lymphadenectomy (eLND) were retrieved from the TCGA database and divided into experimental (*n* = 199) and validation cohorts (*n* = 299). Nodal involvement was chosen as a primary endpoint, whereas other adverse pathological features, including EPE and high-grade cancer (prognostic group III and higher; HGPC), constituted secondary endpoints. Screening selection was based on outcomes of univariate logistic regression and yielded expression of 6 proteins–integrin alpha-2 (CD49b), e-cadherin, heregulin, neurofibromin 2 (NF2), phosphatase and tensin homolog (PTEN), and transferrin receptor 1 (TfR1), predicting NI both in development and validation subsets. Multivariate logistic regression confirmed RPPA quantitative expression of NF2, CD49b, TfR1, and PTEN as independent predictors of NI with an accuracy of exclusive expression data, exceeding 70% (AUC 0.73).

### 2.2. Immunohistochemical Validation

In the second step, the validation cohort was randomly selected from patients who underwent RP and eLND in the years 2012–2018 in a single tertiary center. 

Since archived biopsy specimens were insufficient to achieve optimal staining, postprostatectomy specimen-derived expression patterns were used as substitutes for biopsy expression to simulate a pre-prostatectomy environment. 

Postprostatectomy specimens were retrieved, and hematoxylin and eosin (H&E)-stained slides were examined by an experienced pathologist to choose the most representative slides, containing acinar adenocarcinoma of the prostate. Samples were stained with antibodies against CD49b, e-cadherin, NF2, PTEN, and TfR1, in accordance with the manufacturers’ recommendations. Immunohistochemical staining optimization was achieved for all proteins, except heregulin, which was excluded from further analysis. Slides were then digitalized with a Hamamatsu NanoZoomer 2.0-HT scanner and viewed using NDP.view2 software. Immunohistochemical expression was evaluated semi-quantitatively at 20× magnification as follows: loss or reduced of PTEN and e-cadherin expression in >10% of cancer cells in comparison with positive internal control in normal prostate glands and nerves was scored as PTEN-reduced; loss of e-cadherin expression or faint, incomplete membrane staining in >10% of cancer cells in comparison to positive internal control in normal prostate glands was scored as e-cadherin-reduced; loss of CD49b expression or reduced expression in >25% of cancer cells in comparison to internal positive control in myoepithelial cells of normal prostate glands was scored as CD49b-reduced. NF2 and TfR1 expression in >75% of cancer cells were scored as NF2-positive and TfR-positive, and expression in 25–75% of cancer cells as heterogenous expression and expression in <25% of cancer cells as negative.

### 2.3. Statistical Analysis

Analyses were performed using SAS 9.4 software (SAS Institute, Cary, NC, USA). Continuous and qualitative variables were compared utilizing Mann–Whitney’s U-test and the Fisher’s exact test, respectively. Logistic regression models were utilized for multivariate analysis issues. The threshold for significance was set at *p* < 0.05.

## 3. Results

Results of screening univariate and multivariate logistic regression conducted on TCGA validation cohort are presented in Table 1. 

A total of 53 postprostatectomy specimens were then used for immunohistochemistry validation of preselected candidate proteins. Baseline clinical characteristics of the validating cohort are presented in Table 2, whereas available postprostatectomy characteristics of the TCGA compared to validating cohort are presented in Table 3.

### 3.1. Expression Patterns vs. Adverse Pathological Features

Examples of TFR, e-cadherin, and CD49 immunoreactivity in PCa are presented in Figure 1. Positive expression of transferrin receptor (TFR1) was associated with a higher risk of high-graded cancer (17/20 in patients with HGPC vs. 21/33 in patients with non-HGPC, *p* = 0.0023), whereas PTEN (*p* = 0.53), NF2 (*p* = 0.93), e-cadherin (*p* = 0.99) and CD49b (*p* = 0.38) status failed to achieve statistical significance. The quantitative difference in mean expression values of TfR1 has also shown a trend towards statistical significance (*p* = 0.095, Figure 2).

Reduced expression of CD49b was significantly more common in patients with NI (5/7 71.4% vs. 12/46 26.1%, *p* = 0.028). There was a trend towards significance for the association of reduced e-cadherin expression and NI (2/7 28.6% vs. 3/46 6.5%, *p* = 0.12), whereas when analyzed quantitively there was an association of lower expression levels of e-cadherin with NI (*p* = 0.01, Figure 2). PTEN loss, as well as positive TfR1 status, was more common in patients with NI, but the association failed to prove statistically significance (3/7 42.9% vs. 12/46 26.1%, *p* = 0.39; 4/7 57.1% vs. 18/46 39.1%, *p* = 0.41, respectively). 

Positive expression of transferrin receptor (TfR1) was also associated with a significantly higher risk of extraprostatic disease (16/28 57.1% vs. 34% 6/25, *p* = 0.045). The association has also been visible in quantitative analysis (*p* = 0.075). Reduced expression of CD49b was more common in patients with EPE, although the association failed to prove statically significant (11/28 39.3% vs. 6/25 24%, *p* = 0.26). PTEN and NF2 status, as well as expression of e-cadherin, was not associated with EPE. We failed to optimize heregulin staining; thus, it was not included in further analysis.

### 3.2. Multivariable Analysis

Utilizing post-PR specimen expression patterns that were associated with adverse pathological features in univariate analysis, multivariable regression models were developed for prediction of EPE and NI, respectively (Table 4). Positive expression of TfR1, reduced expression of e-cadherin, and baseline serum PSA level constituted independent predictors of extraprostatic disease, whereas reduced expression of CD49b, reduced expression of e-cadherin, and PSA were independent predictors of nodal involvement. Both models revealed great accuracy (AUROC 0.879 and 0.888, respectively).

## 4. Discussion

In this study, we presented immunohistochemical validation of the preselected protein markers of adverse pathologic features of prostate cancer. The concept of the search for simple tools that can aid pre-prostatectomy decision-making arises from the necessity of patient-tailored extended lymphadenectomy and nerve-sparing. To date, several preoperative nomograms prognosing pN1 and pT3 prostate cancer have been developed and successfully validated [5,6,7]. Most recently, these tools have been supplemented with radiological data from multiparametric magnetic resonance [8,9,10,11,12,13]. The rationale for updating novel nomograms with an even wider variety of variables comes from persistently imperfect accuracy achieved in external validations by the best-calibrated models (74–79% for Briganti nomogram) [8,11], and their troublesome clinical implementation, especially considering approximately 44% eLNDs being performed unnecessary [11]. 

After its validation as clinically feasible in needle biopsy specimen [14,15], immunohistochemistry (IHC) was introduced as a useful adjunct, when predicting adverse pathologic features [15,16,17,18] and treatment outcomes in patients undergoing RP [19,20]. Implementing IHC in a multiparametric MRI-targeted biopsy setting might optimize selecting the sample for staining utilizing index lesion [21]. Considering cost-effectiveness, straightforward implementation and accuracy of IHC, the only practical issue remaining to be solved is the selection of IHC markers that can be clinically confirmed as supplementary to previous tools. 

PTEN and NF2 are well-known cell-cycle regulators, frequently mutated in a wide range of neoplasms. In PCAa, PTEN is recognized as the most commonly inactivated tumor suppressor gene [22]. PTEN loss has been associated with ERG rearrangement [22] and adverse clinical course including post-PR upgrading [15], biochemical recurrence [23], progression after adjuvant treatment [24], as well as cancer-specific death in both hormone-naïve/hormone-sensitive [25], and castration-resistant patients [26]. PTEN status has been confirmed as predictive of 10-year outcomes for mPCAa independently from age, Gleason score, and stage [19], although supplementing the clinical model with PTEN expression yielded a moderate improvement of accuracy (0.76 to 0.8). Neurofibromin 2, a cytoskeletal protein that also belongs to the tumor suppressor group of genes, regulates cellular growth by direct contact. To date, NF2 expression has not been linked with prostate cancer development and prognosis, although in vitro studies suggest that merlin can be inactivated in PCAa cells [27]. Although TCGA analysis indicated both RPPA of PTEN and NF2 as associated with nodal involvement, we failed to confirm IHC expression of either of proteins as a predictor of pN1 or EPE.

Expression of CD49b (also known as integrin alpha2) and e-cadherin, both facilitating adhesion and transducing signals, are commonly dysregulated in the microenvironment of the solid tumors, including prostate cancer. In vitro studies suggest that blockade of integrin alpha2, a putative PCAa stem cell marker, results in poor adhesion to collagen I and inhibition of invasion, whereas its clustering and reorganization may act pro-invasive leading to activation of matrix metalloproteinases [28]. Outcomes of another preclinical study propose interactions of integrin alpha2 with collagens as key molecular event accounting for PCAa metastasis [29]. The following observation was introduced into a clinical scenario with high expression of alpha 2, and alpha 6 integrins, predicting longer metastasis-free survival [30]. By analogy, loss of e-cadherin expression led to worse clinical outcomes. E-cadherin, a prime mediator of cellular cohesion plays a key role in preventing cancer cells from spreading. Matrix metalloproteinase to e-cadherin expression ratio exhibited a strong association with non-organ-confined disease and predicted extracapsular extension independently from biopsy Gleason score and PSA [31]. To date, downregulation of alpha-catenin, which is an intracellular element of e-cadherin, has been suggested as a predictor of unfavorable outcome including baseline grading [31] and staging [31,32,33], positive surgical margins [33], BCR [34], as well as cancer-specific survival in organ-confined [34], locally-advanced [35], and metastatic [32] patients. Existing evidence corresponds with our data. In our study, both adhesion molecules have been confirmed as predictive of NI, with CD49b being additionally significantly associated with EPE. Noteworthy expression of adhesion molecules predicted APFs independently from PSA, with multivariable models reaching almost 0.9 AUROC, suggesting promising potential for clinical implementation. It is, however, worth noticing that expression of both molecules corresponded with EPE or NI in qualitative analysis only, with no significant associations found in quantitative analysis. This outcome might suggest that it is solely an event of loss of adhesion molecules that takes place even in small subpopulation of cancer cells that can drive local and nodal spread.

The last of the candidate proteins provided by TCGA analysis was transferrin receptor 1. Since prostate cancer shows significantly increased iron uptake, presenting thus upregulation of transferrin receptor [36], TfR1 has been early introduced as a marker of transformed prostate cell phenotype [37]. Noteworthy, among the plethora of cellular effects, transferrin itself, has also been suggested to regulate e-cadherin and beta-catenin in prostate cancer cells [38]. TfR1 has not been to date associated with particular clinical outcomes. However, based on our data, TfR1 expression might be a marker of high-graded PCAa and TfR1 status is associated with EPE independently from PSA levels and integrin alpha2 expression.

In this study, we present a set of consecutively developed multivariable models that utilize clinical data and predefined IHC markers to predict post-prostatectomy adverse pathological features. The quasi-internal validation of the models suggests high diagnostic accuracy whereas its clinical implementation appears to be low-cost and straightforward. It can be expected that further studies on larger cohorts might yield more complex models with additional clinical and radiological variables supplementing IHC markers. Further evaluation should also bring external validation of TfR1, e-cadherin, and CD49 immunohistochemical expression depicting its real value in a clinical setting.

The study has several limitations that can be attributed to its retrospective design and limited sample size. The quality and amount of archived biopsy specimens were not sufficient for IHC staining, which forced us to substitute it with post PR specimens. Although the observed correlations seemed unequivocal, the IHC staining would require a check for technical feasibility in a biopsy setting. Finally, a limited number of patients included in the analysis has probably resulted in simplifying the model and prevented us from performing the external validation, as well as head-to-head comparisons with currently used tools.

## Figures and Tables

**Figure 1 jcm-10-05587-f001:**
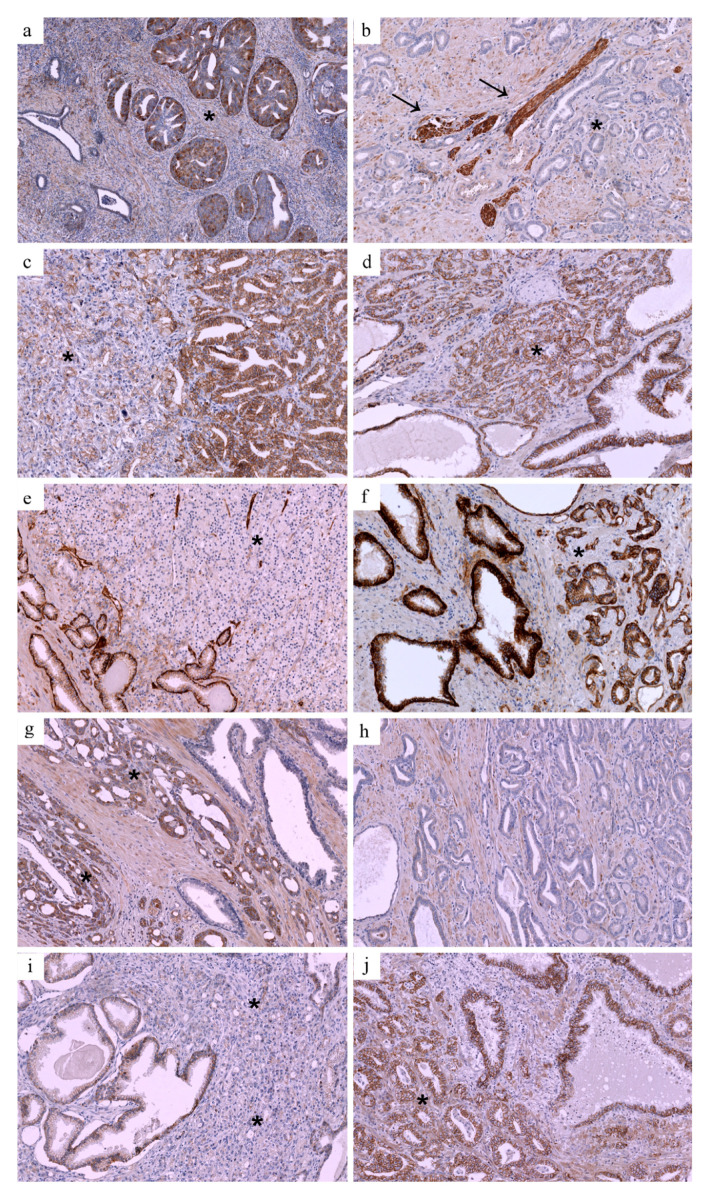
Examples of TFR, e-cadherin, and CD49 immunoreactivity in PCa. (**a**) High-grade PCa with positive TFR expression (marked with asterix); benign prostate glands are visible on the left and serve as internal negative control. (**b**) Low-grade PCa with negative TFR expression; TFR-positive nerve bundles are visible in the center and serve as internal positive control (marked with arrows, cancer cells marked with asterix). (**c**) PCa with NI and reduced e-cadherin expression in high-grade cancer glands on the left (marked with asterix) and normal expression in low-grade cancer glands on the right. (**d**) PCa with no NI and normal e-cadherin expression (marked with asterix). (**e**) PCa with reduced CD49 expression (marked with asterix); benign prostate glands are visible on the left and serve as internal positive control. (**f**) PCa with normal CD49 expression (marked with asterix). (**g**) PCa with EPE and positive TFR expression (marked with asterix); benign prostate glands are visible on the right and serve as internal negative control. (**h**) PCa confined to prostate and negative TFR. (**i**) PCa with EPE and reduced e-cadherin expression in cancer glands on the right (marked with asterix) and normal expression in benign glands on the left. (**j**) PCa confined to prostate with normal e-cadherin expression (marked with asterix). Original magnification in photomicrograph (**a**): 40×, in photomicrographs (**b**–**j**): 100×.

**Figure 2 jcm-10-05587-f002:**
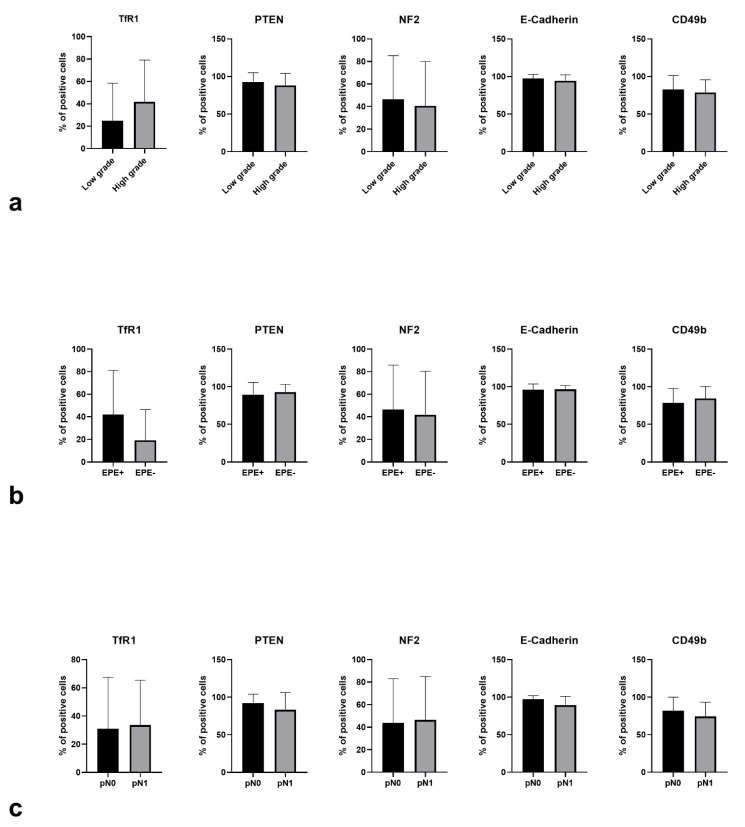
Quantitative (% of positive cells) TfR1, PTEN, NF2, e-cadherin, and CD49b immunoreactivity in (**a**) low-grade vs. high-grade PCa, (**b**) EPE+ vs. EPE- PCa, (**c**) pN0 vs. pN1 PCa. EPE—extraprostatic extension; PCa—prostate cancer; pN0—no nodal involvement; pN1—nodal involvement; TfR1—transferrin receptor 1; PTEN—Phosphatase and tensin homolog; NF2—neurofibromin 2; CD49b—integrin alpha-2.

**Table 1 jcm-10-05587-t001:** Univariate and multivariate analysis of RPPA outcomes for prediction of nodal involvement—TCGA validation subset (*n* = 299) *.

	Univariate	Multivariate	
Proteins preselected from TCGA development subset *	*p*	OR (95% CI)	*p*
CD49b	0.0001	0.06 (0.01–0.41)	0.0008
E-Cadherin	0.0036		
Heregulin	0.0274		
NF2	<0.0001	0.17 (0.04–0.67)	<0.0001
PTEN	0.0015	0.50 (0.27–0.94)	0.0287
TfR1	0.0001	1.83 (1.11–3.03)	0.0137

OR—odds ratio; PSA—prostate specific antigen; CI (confidence interval); CD49b—integrin alpha-2; NF2—neurofibromin 2; PTEN—Phosphatase and tensin homolog; TfR1—transferrin receptor 1. * development subset included *n* = 199 cases whereas validation subset presented in the table included *n* = 299 cases.

**Table 2 jcm-10-05587-t002:** Baseline characteristics of the cohort used for immunohistochemistry validation.

Variable		Number (%)
	<10 ng/mL	26 (49.06%)
PSA	10–20 ng/mL	19 (35.85%)
	>20 ng/mL	8 (15.09%)
	I	14 (27.45%)
	II	15 (29.41%)
Biopsy prognostic group	III	12 (23.53%)
	IV	8 (15.69%)
	V	2 (3.92%)
	T1c	21 (41.18%)
	T2a	19 (37.25%)
cT	T2b	4 (7.84%)
	T2c	6 (11.76%)
	T3	1 (1.96%)

PSA—prostate-specific antigen; cT—clinical stage.

**Table 3 jcm-10-05587-t003:** Postprostatectomy characteristics of TCGA cohort and validation cohort used for immunohistochemistry.

Variable		Number (%)		*p*
		TCGA cohort	Validation cohort	
Postprostatectomy prognostic group	I	44 (8.87%)	3 (5.66%)	0.22
	II	148 (29.84%)	16 (30.19%)	
	III	101 (20.36%)	14 (26.42%)	
	IV	64 (12.9%)	11 (20.75%)	
	V	139 (28.02%)	9 (16.98%)	
Nodal involvement		79 (18.63%)	7 (13.21%)	0.44
Extracapsular extension		158 (31.79%)	14 (26.42%)	0.53
Seminal vesicle invasion		134 (26.96%)	14 (26.42%)	0.99
Positive surgical margins		151 (36.26%)	26 (49.06%)	0.038

**Table 4 jcm-10-05587-t004:** Multivariate analysis for prediction of extraprostatic disease (a), and nodal involvement (b).

**(a) Extraprostatic Disease (AUROC = 0.879)**			
**Variable**		**OR (95% CI)**	** *p* **
TfR1 status	Heterogenic	7.54 (0.95–59.65)	0.015
	Positive	13.74 (1.48–127.54)	
CD49b status	Reduced	10.15 (1.37–75.43)	0.013
PSA (ng/mL)		1.29 (1.11–1.51)	0.013
**(b) nodal involvement (AUROC = 0.888)**			
e-cadherin status	Reduced	10.22 (0.74–142.11)	0.005
CD49b status	Reduced	24.44 (0.79–756.37)	0.017
PSA (ng/mL)		1.18 (0.96–1.45)	0.002

AUROC—area under ROC curve; OR—odds ratio; PSA—prostate specific antigen; CI (confidence interval); TfR1—transferrin receptor 1; CD49b—integrin alpha-2.

## Data Availability

The datasets used and analyzed during the current study are available from the corresponding author on reasonable request.

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
