# Peer review of "E-Cadherin, Integrin Alpha2 (Cd49b), and Transferrin Receptor-1 (Tfr1) Are Promising Immunohistochemical Markers of Selected Adverse Pathological Features in Patients Treated with Radical Prostatectomy"

_jcm, 2021, doi:10.3390/jcm10235587_

Round 1
Reviewer 1 Report
The manuscript is generally well written. The title is informative and reflects the content, and the abstract is clear and complete.
Some minor comments listed below would make the manuscript sound clearer:
- Why is heregulin excluded from the further analysis? Because of no statistical significance (p=0,0274) from univariate analysis? It should be mentioned (in the section Immunohistochemical validation)
- Table 1. n=199 in the table and in the table title it stands that n=299. Is it a mistake?
- Result and discussion – well written in general
Author Response
The manuscript is generally well written. The title is informative and reflects the content, and the abstract is clear and complete.
We are sincerely grateful for this kind opinion of the Reviewer!
Some minor comments listed below would make the manuscript sound clearer:
Why is heregulin excluded from the further analysis? Because of no statistical significance (p=0,0274) from univariate analysis? It should be mentioned (in the section Immunohistochemical validation)
Thank you for precepting this. Indeed we needed to exclude the heregulin from further analysis despite its significance in TCGA. The reason for this was the fact we could not optimize staining with heregulin antibodies. In the previous manuscript version, it was shortly stated in the material and methods section but since a single mention can be easily missed we have supplemented the results with the clarifying sentence.
Table 1. n=199 in the table and in the table title it stands that n=299. Is it a mistake?
Thank you for noticing this inaccuracy. As mentioned in methods we divided TCGA cohort (n=498) onto development / experimental cohort (n=199) and validation cohort (n=299). We wanted this to be pointed out in the table that the proteins presented as predictors in regression analysis came from analysis performed in the development cohort which counted n=199. Since however, it might be misleading and unclear we changed the explanation and provided additional comment under the table to avoid doubts.
Result and discussion – well written in general
We are sincerely grateful to the Reviewer for recognizing our work as valuable and hope our manuscript to be found interesting to all JCM Readers.
Reviewer 2 Report
CRITIQUES:
- Authors need to include clinical characteristics of patients used for experimental cohort for TCGA analysis, as well as a side-by side comparison of discovery and validation cohorts.
- Result section 3.1 needs to include quantitative graphs for each marker comparing high vs. low-grade cancers +/- NI. This is especially important as markers like E-cadherin whose reduced expression was not a significant marker via IHC staining but was significant as a marker for NI through simulation.
- For result section 3.2, how was the simulation performed? Details need to be added to the Methods section. Need supporting IHC data for showing utility of E-cadherin for NI.
- Since some markers, for example E-cadherin, have different prognostic value depending on cancer grade or NI, the specifics need to be separately stated in the abstract and not in a blanket statement.
Author Response
Authors need to include clinical characteristics of patients used for experimental cohort for TCGA analysis, as well as a side-by side comparison of discovery and validation cohorts.
Thank you for this rational suggestion. Obviously, the comparison between development and validating cohorts is worth presenting considering the risk of bias arising from differences in datasets. We supplemented the manuscript with an additional table (Table 3) moving some of the content from Table 2. The comparison included what was the main outcome of the study so adverse pathological features and postprostatectomy grading. Unfortunately, some particular clinical data (PSA and biopsy grade group) were not available from TCGA dataset and thus were not presented. Clinical staging (DRE) was available, but considering its subjectivity, poor accuracy for the prediction of T3 and since it could be considered rather insignificant for further results, we decided not to present it, but rather move it to a supplementary file.
Please also find modified Figure 1 supplemented with additional pictures comparing EPE+/-, NI+/- and HG+/- as well as a modified description of Figure 1.
Result section 3.1 needs to include quantitative graphs for each marker comparing high vs. low-grade cancers +/- NI. This is especially important as markers like E-cadherin whose reduced expression was not a significant marker via IHC staining but was significant as a marker for NI through simulation.
Thank you for this remarkable suggestion. Please find section 3.1 supplemented with new Figure 2 with graphs depicting quantitive expression comparisons for all three end-points. Since the loss of expression in some of the markers (especially e-cadherin) is informative in case of any % of cells losing it (5-10%) qualitative analysis shows significant differences even though quantitive analysis shows mild or no differences. It means it is rather a matter of the event, not its intensity. Basically, we interpret this observation as a consequence of only a small subgroup of cells requiring particular mutations to start spreading. This is probably why even small groups of cancerous cells losing E-cadherin (resulting in a small quantitive change in total expression) is a significant event in patients with nodal involvement. To elucidate the issue properly we have also supplemented the discussion with two sentences commenting on this outcome.
For result section 3.2, how was the simulation performed? Details need to be added to the Methods section. Need supporting IHC data for showing utility of E-cadherin for NI.
Thank you for pointing out this inaccuracy. We have previously mentioned that since it was not possible to use archived biopsy specimens (it was simply a matter of a small amount of tissue) our aim was to simulate a pre-prostatectomy environment utilizing a postprostatectomy specimen. Since we see this as a limitation (it would be best if we were able to check the expression in biopsy cores, in fact, the prospective validation in biopsy cores is being launched within upcoming weeks) we used the word “preoperative simulation” just to make the reader aware of this imperfection. We agree this is somewhat misleading and we simply changed paragraph 3.2 and provide some more direct description within the section method. We have also supplemented the IHC description within the methods section.
Since some markers, for example E-cadherin, have different prognostic value depending on cancer grade or NI, the specifics need to be separately stated in the abstract and not in a blanket statement.
Thank you for this accurate suggestion. We hope we followed your intention correctly and changed two sentences describing outcomes of analysis to stress the fact that it was not all adverse pathological features that chosen markers managed to predict but it was outcome dependent (including E-cadherin being predictive only of NI and TfR1 being predictive only of EPE). To keep the title informative and citable we kept the detailed information within it but stressed the specificity of association.
Round 2
Reviewer 2 Report
Authors have addressed all the major concerns.